# Application of FTIR Spectroscopy to Detect Changes in Skeletal Muscle Composition Due to Obesity with Insulin Resistance and STZ-Induced Diabetes

**DOI:** 10.3390/ijms232012498

**Published:** 2022-10-18

**Authors:** Barbara Zupančič, Nejc Umek, Chiedozie Kenneth Ugwoke, Erika Cvetko, Simon Horvat, Jože Grdadolnik

**Affiliations:** 1Laboratory for Molecular Structural Dynamics, Theory Department, National Institute of Chemistry, 1000 Ljubljana, Slovenia; 2Institute of Anatomy, Faculty of Medicine, University of Ljubljana, 1000 Ljubljana, Slovenia; 3Chair for Genetics, Biotechnology and Immunology, Biotechnical Faculty, University of Ljubljana, 1230 Domžale, Slovenia

**Keywords:** Fourier transform infrared (FTIR) spectroscopy, histochemical analysis, insulin resistance, obesity, diabetes, skeletal muscle macromolecular composition

## Abstract

Age, obesity, and diabetes mellitus are pathophysiologically interconnected factors that significantly contribute to the global burden of non-communicable diseases. These metabolic conditions are associated with impaired insulin function, which disrupts the metabolism of carbohydrates, lipids, and proteins and can lead to structural and functional changes in skeletal muscle. Therefore, the alterations in the macromolecular composition of skeletal muscle may provide an indication of the underlying mechanisms of insulin-related disorders. The aim of this study was to investigate the potential of Fourier transform infrared (FTIR) spectroscopy to reveal the changes in macromolecular composition in weight-bearing and non-weight-bearing muscles of old, obese, insulin-resistant, and young streptozotocin (STZ)-induced diabetic mice. The efficiency of FTIR spectroscopy was evaluated by comparison with the results of gold-standard histochemical techniques. The differences in biomolecular phenotypes and the alterations in muscle composition in relation to their functional properties observed from FTIR spectra suggest that FTIR spectroscopy can detect most of the changes observed in muscle tissue by histochemical analyses and more. Therefore, it could be used as an effective alternative because it allows for the complete characterization of macromolecular composition in a single, relatively simple experiment, avoiding some obvious drawbacks of histochemical methods.

## 1. Introduction

The triad of aging, obesity, and diabetes mellitus (DM) is arguably the most important contributing factor for the non-communicable disease morbidity and mortality burden in the modern world. All three conditions are associated with a broad spectrum of alterations in metabolic phenotype, with dysfunction in insulin action or secretion being one of the core pathophysiological common denominators. Skeletal muscles are critically involved in the modulation of carbohydrate, lipid, and protein metabolism, and undergo a range of structural and functional changes in the setting of defective or deficient insulin signalling. They are composed of distinct fibre types with different physiochemical and metabolic properties [1,2,3]. Depending on their oxidative and glycolytic capacity, healthy skeletal muscles can rapidly switch between carbohydrate and lipid fuels in accordance with the bioenergetics demands, and the loss of this flexibility in fuel selection is one of the hallmarks of insulin resistance associated with aging, obesity, and type 2 DM [4,5].

The biomolecular skeletal muscle changes associated with type 1 DM are expected to differ from those seen in type 2 DM [6] and related metabolic phenotypes such as obesity with insulin resistance and aging due to the different aetiopathogenesis of the conditions. For example, while elevated intramyocellular lipids have been extensively described in type 2 DM and similar insulin-resistant conditions [7,8], studies of the type 1 DM phenotype have reported both increased and unchanged intramyocellular lipids [9,10]. Aging is associated with progressive changes in body composition, and notably the accumulation and redistribution of fat (including ectopic deposition of fat in the abdomen, liver, skeletal muscles, etc.) and loss of lean mass [5]. In addition to senile sarcopenia and intramyocellular lipid accumulation during skeletal muscle aging, several other mechanisms have been identified (e.g., chronic inflammation, hyper-activation of the renin–angiotensin system, enhanced reactive oxygen species (ROS) production, oxidative stress, endoplasmic reticulum stress, decline in autophagy capacity, and altered activity of enzymes regulating insulin sensitivity), all of which contribute to an increased risk of developing insulin resistance, type 2 DM, and cardiovascular disease [4,5]. Meanwhile, DM exacerbates the decline in lean mass with aging [5]. Chronic obesity and type 2 DM are similarly associated with ectopic lipid infiltration in non-adipose tissues, notably skeletal muscles. Such accumulation in total lipids and specific lipid species in skeletal muscles and their localization within the cells are associated with changes in tissue composition, architecture, and remodelling, as well as consequent metabolic dysfunctions including insulin resistance and reduced glucose uptake, lipotoxicity, and mitochondrial dysfunction [11,12]. In addition, Masgrau et al. showed that increased lipid accumulation in glycolytic muscles is associated with a decreased protein synthesis rate seen in the later phase of obesity development [12].

While some studies have investigated diagnostic biomarkers for pre-diabetic phenotypes [13,14], it remains essential to identify additional biomarkers and their combinations that are more sensitive and accurate in predicting the metabolic spectrum and progression of impaired insulin action. This is necessary to mitigate or even prevent long-term complications. Therefore, the changes in skeletal muscle microanatomy, particularly the alterations in macromolecular composition, structure, and dynamics in the evolution of types 1 and 2 DM phenotypes remain an active focus of investigation. While it is known that different metabolic pathways of specific muscle fibre types govern the way myofibres are affected by these metabolic conditions [15], several inconsistencies and uncertainties remain in understanding the effects of these conditions on the content, breakdown, and/or synthesis of lipids and proteins in specific skeletal muscle types [11].

In recent years, Fourier transform infrared (FTIR) spectroscopy has been gaining increasing recognition as an effective technique for determining macromolecular changes in biological tissues [16]. The high molecular specificity and the rapid and automated measurement with minimal sample preparation effort support the diagnostic potential of FTIR spectroscopy in medicine. It is arguably superior to the widely used traditional histochemical techniques [17], which are time-consuming and require extensive sample preparation protocols with changes in native tissue structure. The FTIR spectrum contains information about the characteristic vibrational bands that help to identify the types of macromolecules present in the tissue [18]. It allows for the simultaneous highly sensitive detection of changes in the functional groups of biomolecules and can therefore indicate subtle alterations in tissues [19].

Only a few studies have used FTIR spectroscopy to characterize tissue components in skeletal muscle in a small number of pathological contexts, such as drug-induced myotoxicity, juvenile-onset obesity, and streptozotocin (STZ)-induced type 1 DM [15,19,20]. In the present study, we primarily aimed to investigate the ability of FTIR spectroscopy with accompanying chemometrics in detecting differences in skeletal muscle macromolecular composition as observed by histochemical analysis and, accordingly, to evaluate its potential for clinical application. Weight-bearing and non-weight-bearing muscles from old, obese, insulin-resistant, and young streptozotocin (STZ)-induced diabetic mice, as well as the corresponding control groups, were used for the study. Due to the sensitivity to various abnormalities described above, an accurate characterization of the biomolecular composition of skeletal muscle using FTIR spectra not only provides a better understanding of the molecular basis of various metabolic diseases, but can also provide an accessible and relatively simple approach for disease staging, phenotypic classification, prognosis, monitoring of disease progression, and response to treatment in various myopathies and metabolic diseases.

## 2. Results

### 2.1. Animals

The young STZ-induced diabetic mice (Y-STZ-DM) had a significantly lower body mass in comparison to young non-diabetic controls (Y-NDM) and old non-diabetic controls (O-NDM). In contrast, old HFD-induced obese insulin-resistant mice (O-HFD-DM) had a significantly higher body mass than young and old non-diabetic controls, between which no significant difference in body mass was observed (Table 1). STZ-induced diabetic mice developed severe type 1 DM with markedly elevated fasting glucose levels that were statistically significantly increased compared to young non-diabetic, old non-diabetic, and old HFD-induced obese, insulin-resistant mice. HFD-induced obese mice developed moderate insulin resistance with basal hyperglycaemia (Table 1) and decreased tolerance for glucose characteristic of type 2 prediabetic states (Figure 1).

### 2.2. Muscle Compositional Changes Analysed by FTIR Spectroscopy

The results of FTIR spectroscopy are shown in Figure 2, where the solid black line in each graph represents the average spectrum of *n* = 9 normalized spectra for each of the four study groups and the two muscles studied, and the grey shaded area indicates the range between the minimum and maximum spectral intensity at the corresponding wavenumber. The plots on the left show the spectra of the non-diabetic control groups, while the right plots show the spectra of the STZ-induced diabetic group and the insulin-resistant obese group. It is evident that the grey shaded area for the gluteus maximus muscle spans a wider range of intensities compared to the gastrocnemius muscle, suggesting that the variability in gluteus maximus composition within the group is more pronounced than the variability in gastrocnemius muscle composition.

After MCR decomposition of the FTIR spectra into the two spectral components, components *SC*1 and *SC*2 were obtained. The two are shown in Figure 3a. The corresponding weights *c*1 and *c*2 for the spectral components are shown in Figure 3b for each specimen within a given group and for a given muscle.

Using the second derivatives of the components *SC*1 and *SC*2 obtained from the MCR decomposition, we were able to identify the wavenumbers of the band peaks. They are listed in Table 2, along with the spectral assignments.

From a detailed examination of the spectral bands in Table 2, it is evident that the two spectral components *SC*1 and *SC*2 from the MCR decomposition provide a clear separation between the macromolecular constituents of muscle tissue based on the characteristic absorption bands specific to each spectral component. Based on the assigned bands, it can be determined that *SC*1 is mostly a lipid-dominated component and *SC*2 is a protein-dominated component. *SC*1 contains mainly information on unsaturated fatty acids (absorption band with the peak at 3007 cm^−1^, see Table 2, *SC*1), fatty acid esters, and phospholipids (absorption bands with the peaks at 1743, 1377, 1361, 1343 cm^−1^, etc., see Table 2, *SC*1), with less pronounced but still present evidence of glycogen (absorption bands with the peaks at 1082, 1023 and 767 cm^−1^, see Table 2, *SC*1). On the other hand, the most intense bands of *SC*2 belong to proteins (amide A, B, I, II, and III absorption bands, see Table 2, *SC*2) with specific peaks of collagen and glycated collagen (absorption bands with the peaks at 1340, 1241, 1081, and 1041 cm^−1^, see Table 2, *SC*2) and tyrosine (absorption bands with the peaks at 1513, 849, and 830 cm^−1^, see Table 2, *SC*2), while the remaining peaks mainly denote nucleic acids (absorption bands with the peaks at 1122, 1103, 972, 929, 792, 779, and 772 cm^−1^, see Table 2, *SC*2). Therefore, the weight *c*1 defines the portion of lipids and glycogen, while the weight *c*2 defines the portion of proteins and nucleic acids. From the data on the two weights, it can be seen that within the group, the protein-dominated component is more scattered than the lipid-dominated one for the gluteus maximus muscle (Figure 3b—upper plots), while it is the opposite for the weight-bearing gastrocnemius muscle (Figure 3b—lower plots). Nevertheless, as already evident from the raw FTIR spectra, the overall scatter of the *c*1 and *c*2 weights within the groups is less pronounced for the gastrocnemius muscle than for the non-weight-bearing gluteus maximus.

It is also important to highlight the peculiarity observed in the MCR spectral components in Figure 3a and summarized with the band assignments in Table 2, i.e., the presence of amide I and amide II bands of α-helical and β-sheet protein secondary structures not only in the *SC*2 (protein-dominated) component but also with a smaller contribution in the *SC*1 (lipid-dominated) component. This result indicates the changes in the proportions of certain secondary structures, i.e., the changes in the overall pattern of protein composition controlled by both *c*1 and *c*2 weights.

Statistical consideration of *c*1 and *c*2 has further revealed how we can distinguish between groups and the two muscle types within the group (Figure 4).

It appears that there is greater variability between the groups and between muscles in the same group for the lipid-dominated component (Figure 4a) than for the protein-dominated component (Figure 4b). From Figure 4, we can also conclude that the macromolecular composition of the gluteus maximus is significantly different from that of the gastrocnemius muscle in all study groups. If we combine the results shown in Figure 4 for the two weights that determine the contribution of each of the two spectral components with the band assignments given in Table 2, we can summarize that a statistically significant increase (decrease) in weight *c*1 means a statistically significant increase (decrease) in unsaturated fatty acids, fatty acid esters (mostly triglycerides), phospholipids, and glycogen when we compare between two study groups or two muscles. Similarly, we can summarize that a statistically significant increase (decrease) in weight *c*2 means a statistically significant increase (decrease) in proteins, collagen, glycated collagen, tyrosine, and nucleic acids. When a statistically significant difference is found in at least one of the *c*1 and *c*2 weights when comparing two study groups or two muscles, it also means that the patterns of overall protein composition have changed due to the presence of amide I and amide II bands of α-helical and β-sheet protein secondary structures, which were detected not only in the *SC*2 component but also with a smaller contribution in the *SC*1 component. The findings elaborated above are summarized in Table 3.

Bozkurt et al. applied the FTIR technique to investigate the macromolecular changes in skeletal muscles of the STZ-induced type 1 diabetic rat and found that DM induced several compositional and structural alterations, including a decrease in lipids (phospholipids, fatty acids, and triglycerides), protein and nucleic acid contents, an increase in membrane order and fluidity, modifications of protein secondary structure, and reduced integrity of collagen molecules [15]. Similarly, Toyran et al. [19] used FTIR to investigate the effect of type 1 DM on the macromolecular composition not of skeletal muscle but of rat cardiac muscle tissue. They also found alterations in the content of lipids, glycogen, collagen, and secondary structure profile of proteins compared to the control group.

Except for the decreased nucleic acids noted in our FTIR study of skeletal muscle, other findings in the study of Bozkurt et al. [15] are in perfect agreement with our observation for weight *c*1 and consideration of the spectral band at ~3007 cm^−1^ (decrease in the band intensity indicating a loss of unsaturation in acyl chains due to increased lipid peroxidation [15]) for the young STZ-induced diabetic group relative to the young non-diabetic group. In type 1 DM, there are changes in the muscle metabolic properties including altered lipid metabolism, microvascular dysfunction, and oxidative stress [74,75]. The elevated reactive oxygen species due to hyperglycaemia in type 1 DM causes damage to cellular structures and may contribute to the development of insulin resistance [74].

Alterations in phospholipids, which we have noted in the FTIR analysis, are also clinically and experimentally associated with insulin resistance, but it is not clear whether changes in phospholipids are the cause or the consequence of insulin resistance [76]. The higher amount of glycated collagen for the young STZ-induced diabetic group relative to the young non-diabetic group is concordant with the findings of Chiu et al. [77] on diabetic myopathy, and can be related to the activation of the receptor for advanced glycation end product (AGE) by glycated collagen. Changes in the FTIR spectral bands of the *SC*2 component belonging to serine, threonine, and tyrosine residues in proteins denotes the modifications in serine/threonine versus tyrosine phosphorylation relevant to insulin receptor substrate (IRS)-1 [78,79].

### 2.3. Muscle Compositional Changes by Histochemical Analysis

The results of the histochemical analysis of the macromolecular compositional changes of the gluteus maximus and gastrocnemius muscles are summarized in Figure 5, and representative images of histochemically stained sections are shown in Figure 6. The lipid composition of the muscles exhibited the greatest variability across the different study groups. When comparing both muscles, both the old HFD-induced obese insulin-resistant and old non-diabetic mice had significantly higher glycogen and lipid contents (*p* < 0.0001) than the young STZ-induced diabetic mice. The latter showed decreased lipid but increased collagen content relative to the young non-diabetic mice, which compared to the old non-diabetic mice had a lower composition of both macromolecules. Notably, when comparing both muscles, there were no significant differences in the glycogen, lipid, and collagen contents between the old HFD-induced obese insulin-resistant and old non-diabetic mice. Nevertheless, comparisons of the individual muscles revealed further details on the macromolecular compositional changes across the different study groups. Compared to the gastrocnemius muscle, the gluteus maximus muscle expressed higher lipid (except in the young STZ-induced diabetic mice) and collagen content across the groups, and decreased glycogen content in the young STZ-induced diabetic and the young non-diabetic group. In contrast to the gastrocnemius muscle, which did not exhibit a significant difference in the glycogen composition across the study groups, the gluteus maximus of both the old HFD-induced obese insulin-resistant and old non-diabetic mice showed higher glycogen stores compared to the young non-diabetic mice (*p* = 0.0019 and *p* = 0.0074, respectively).

## 3. Discussion

In the present study, we applied FTIR spectroscopy compared with standard histochemical methods to analyse the changes in macromolecular composition of the non-weight-bearing gluteus maximus and weight-bearing gastrocnemius muscles of mice in relation to age, obesity with insulin resistance/type 2 DM, and STZ-induced DM. To evaluate the effectiveness of FTIR spectroscopy compared to the histochemical tissue examination, we reviewed the concordant observation of statistical differences where appropriate. This comparison is shown in Figure 7 for glycogen, lipids, and collagen.

As shown in Figure 7, both experimental approaches resulted in statistical differences in glycogen levels with an increasing trend for the old non-diabetic and the old HFD-induced obese insulin-resistant mice relative to the STZ-induced diabetic group. FTIR spectroscopy also evidenced significantly different glycogen levels in the skeletal muscles of the STZ-induced diabetic group with a decreasing trend compared to the young non-diabetic group, while the histochemical examination did not. However, the observed decrease in glycogen levels for the young STZ-induced diabetic group relative to the young non-diabetic control group from the FTIR spectra is in agreement with [80], in that a deficiency in insulin secretion, as seen in type 1 DM, greatly diminishes the ability of muscle and adipose tissue to store glucose in the form of glycogen. In contrast to our observations (no significant differences in the glycogen content for old HFD-induced obese insulin-resistant relative to old non-diabetic mice), Sullivan et al. [80] reported a decreased amount of glycogen in skeletal muscle in type 2 DM, which may be partly explained by the impaired insulin-mediated glycogen synthase activity for non-oxidative glucose metabolism in skeletal muscle in diabetes [81]. It is worth noting that the HFD-induced obese insulin-resistant mice used in the present study exhibited relatively moderate basal hyperglycaemia and decreased tolerance for glucose, representing more of a pre-diabetic state than advanced diabetes, which may explain the slight discrepancy in results compared to the study of Sullivan et al. [80].

As for the comparison between the two muscles within the group, statistical differences in muscle glycogen were detected for the gluteus maximus relative to the gastrocnemius muscle within the young non-diabetic and the young STZ-induced diabetic groups, but with an opposite trend of the differences obtained in the two experimental approaches (Figure 7—red boxes in the glycogen column). On the other hand, FTIR spectroscopy and histochemical assays yielded inconsistent results on glycogen content in gluteus maximus compared to gastrocnemius muscle for the old non-diabetic and old HFD-induced obese insulin-resistant groups (Figure 7—yellow boxes in the glycogen column). The incongruity in terms of detection of statistical difference probably denotes the superior sensitivity of the FTIR method compared to histochemical analysis in this case.

Histochemical studies revealed that the HFD-induced obese insulin-resistant mice had elevated lipid species compared to the young and old non-diabetic controls, whereas the FTIR spectra showed no statistical differences, suggesting that the latter was less sensitive in detecting changes in this case (Figure 7—yellow boxes in the lipid column). In other comparisons, both approaches gave matching results (Figure 7—green boxes in the lipid column). The marked decrease in muscle lipid content in the STZ-induced diabetic mice compared to the young non-diabetic group may be due to the compensatory switch in fuel selection from glucose to lipids. This switch enhances skeletal muscle lipolysis and depletion of lipid stores as seen in type 1 DM, while the loss of unsaturation in the acyl chains of lipids (see the spectral band at ~3007 cm^−1^) is due to increased lipid peroxidation [15]. Both histochemical analysis and FTIR spectra showed a significantly higher amount of lipid species in the gluteus maximus than in the gastrocnemius muscle.

The most variable comparability between the two experimental approaches was obtained in the analysis of collagen content. This could be partially because collagen content in FTIR spectroscopy was evaluated on the basis of *c*2 weight, which represents the overall portion of proteins and not only collagen. Meanwhile, collagen content in the histochemical analysis was evaluated relatively as area stained for collagen per total cross-sectional area of muscle tissue, which also includes the cross section of the muscle fibres. Nevertheless, we noted that the collagen content was increased in the young STZ-induced diabetic mice compared to the young non-diabetic group. While the histochemical analysis revealed a statistically significant higher collagen content in the old non-diabetic group compared to the young non-diabetic group, this could not be detected from the FTIR results. On the other hand, FTIR spectroscopy detected less collagen in the old non-diabetic group compared to the young STZ-induced diabetic mice, while histochemical assays did not. Nevertheless, based on our findings, we cannot confirm what Berria et al. found, namely that insulin resistance is associated with altered extracellular matrix and increased collagen expression in the skeletal muscle, related to the pathophysiological mechanisms of oversupply and underutilisation of muscle lipids [82].

Meanwhile, it is possible that FTIR spectroscopy detected protein degradation products in addition to collagen. Though very different diseases, type 1 and type 2 DM share mechanisms of high protein turnover, resulting in muscle wasting that is mainly linked to decreased insulin signalling and induction of genes involved in the ubiquitin–proteasome pathway [83]. These activated protein catabolic pathways may have contributed to a high variability in the analysis of collagen content using FTIR spectroscopy. As can be seen from the comparison of the observed statistical differences between gluteus maximus and gastrocnemius collagen content, both experimental approaches are either discordant in terms of the detected trend of changes (Figure 7—the corresponding red boxes in the collagen column) or incongruent in terms of sensitivity to detect changes (Figure 7—the corresponding yellow boxes in the collagen column). One of the possible reasons for this discrepancy might be a widely varying collagen content within and between muscles, which is driven by the complex intramuscular connective tissue distribution throughout a muscle [84].

Our findings also show that the macromolecular composition of skeletal muscles in the different metabolic phenotypes significantly differed according to muscle functional typology. For example, it has been demonstrated that weight-bearing had a significant effect on the expression of myosin heavy chain isoforms and that increased weight-bearing may protect fast-twitch skeletal muscles from fibre-type shifts and associated physiochemical alterations [85,86]. Thus, studies using functionally heterogeneous skeletal muscle types to characterise a particular metabolic phenotype are sometimes not validly comparable, as our results have shown that differences between individual muscle types in terms of the effect studied may be more pronounced than differences between study groups. The different outputs of statistical evaluation of results from FTIR spectroscopy and histochemical analysis could be due to the relatively large variations in tissue composition within each group and would provide a different result if the sample was larger. However, the agreement between the statistical treatments of the results of FTIR spectroscopy and histochemical analysis has a predominant status in Figure 7 (green boxes) and therefore suggests that FTIR spectroscopy is powerful enough to complement or even replace histochemical tests. From the assignment of the FTIR spectral bands, it is evident that the spectroscopic data provide a detailed insight into the composition of the muscle tissue. Furthermore, a single FTIR spectrum contains much more comprehensive information about the tissue constituents than several independent histochemical methods can provide. Another valuable tool in post-processing FTIR spectra is the method for decomposition into the independent spectral components, which can clearly separate the spectral bands belonging to a particular macromolecular substituent throughout the frequency range studied.

Overall, further studies are needed to expand our understanding of the pathophysiological basis and clinical significance of biomolecular changes in different skeletal muscle tissues across the spectrum of metabolic phenotypes related to impaired insulin function. Given that only two skeletal muscle types were examined in female mice in the present study and considering the range of phenotypic variations seen in metabolic disorders, larger studies utilizing both male and female mice or other murine models as well as a wider skeletal muscle typology in a broader metabolic context are warranted to further validate our findings and the role of FTIR spectroscopy in biological tissue analysis.

## 4. Materials and Methods

The Ethical Committee for laboratory animals of the Republic of Slovenia reviewed and approved all study protocols (Permit numbers: U34401-34/2013/6, U34401-34/2014/9). The study was performed in full compliance with the European Union legislation on the protection of animals used for scientific purposes (Directive 2010/63/EU) and in accordance with the recommendations of the National Institutes of Health’s Guide for the Care and the Use of Laboratory Animals and the ARRIVE guidelines.

### 4.1. Animals

Thirty-six female C57BL/6JOlaHsd mice obtained from the Harlan Laboratories Envigo (Desio, Italy) were used in the study. The animals were reared in individually ventilated cage systems at the Centre for Laboratory Animals of the Biotechnical Faculty of the University of Ljubljana, Slovenia. The rearing environmental conditions were as follows: temperature maintained at 23  ±  1 °C; humidity at 40–60%; with 12 h light/dark cycle and water available ad libitum. After one week of acclimatisation, the animals were randomly assigned to four study groups: (a) 13-week old control non-diabetic group (*n* = 9); (b) 13-week old STZ-induced diabetic group (*n* = 9); (c) 54-week old control lean group (*n* = 9); and (d) 54-week old high-fat-diet (HFD)-induced obese insulin-resistant group (*n* = 9) [87]. In the latter group, obesity and insulin resistance were induced at the age of 18 weeks by purified HFD ad libitum for 26 weeks (D12108C, Research Diets, Inc., New Brunswick, NJ, USA). The STZ-induced DM was induced by intraperitoneal injection of 200 mg kg^−1^ STZ at the age of 8 weeks. STZ is an alkylating agent that induces degeneration in pancreatic β islets [88,89]. The control groups and STZ-induced diabetic group received standard maintenance diet ad libitum (4RF18, Mucedola, Milan, Italy).

The status of basal glycaemia of all animals was monitored by fasting glucose levels using Bayer Contour glucose meter (Ascensia Diabetes Care Holdings AG, Basel, Switzerland). The status of insulin resistance in 54-week old groups was monitored using the oral glucose tolerance test (OGTT): a 25% glucose solution (2 g kg^−1^ glucose) was administered via orogastric tube feeding following 6-h fasting (water available ad libitum) [87] with venous blood glucose assayed before, and at 15, 30, 60, and 120 min after the glucose administration. Blood samples were obtained from the tail vein of the animals.

The animals were sacrificed by cervical dislocation at the age of 13 or 54 weeks, following which the gluteus maximus and gastrocnemius muscles were harvested from the left hind limbs, frozen in liquid nitrogen, and preserved at −80 °C until analysis.

### 4.2. FTIR Spectroscopic Analysis and Data Processing

For FTIR spectroscopic analysis, we used several serial transverse 200 μm thick cryosections of the gluteus maximus and gastrocnemius muscles cut with Leica CM 1950 (Leica Microsystems, Wetzlar, Germany) and placed on microscope slides. The 72 specimens thus obtained from the two muscles of 36 animals (one cryosection (specimen) for each of the two muscles from each of the nine individuals within each of the four groups—see the previous subsection) were air-dried at 37 °C for 48 h. The slides with the desiccated muscle tissues were then stored in the refrigerator at 5 °C for 2–3 days before FTIR spectroscopic measurements. Measurements of attenuated total reflection FTIR (ATR-FTIR) spectra were performed at room temperature on the Bruker Tensor 27 spectrometer using an ATR sampling Specac Golden Gate accessory with a single reflection diamond. A nitrogen-cooled MCT detector was used, and the spectrometer optics and ATR cell were sealed from the atmosphere and purged with technical dry nitrogen during measurements. ATR-FTIR spectra in the range of 4000 cm^−1^–600 cm^−1^ were obtained by processing the recorded interferograms with a resolution of 2 cm^−1^ and 64 scans averaged for each spectrum using the OPUS version 7.8 software (Bruker, Billerica, MA, USA). The same software was used to perform atmospheric water and CO_2_ compensation, baseline correction, and min-max normalization (so that the minimum is 0 and the maximum absorbance of the highest peak is 2) for each spectrum, which is subsequently denoted as *_muscle_S_group,i_*. We then subjected the spectra to the multivariate curve resolution-alternating least squares (MCR-ALSs) iterative algorithm using the MATLAB software MCR-ALS GUI v4c [90,91], where the same two spectral components, namely *SC*1 and *SC*2, from the decomposition explained more than 98.3% of the total variance (for more details on MCR, see Appendix A), i.e.,
(1)Smuscle group,i≈cmuscle 1group,i·SC1+cmuscle 2group,i·SC2formuscle∈{gluteus maximus, gastrocnemius}group∈{Y-NDM, Y-STZ-DM, O-NDM, O-HFD-DM}i∈{1,2,…9}


The frequencies of the broad overlapped bands of the two MCR spectral components *SC*1 and *SC*2 were determined via the second derivatives of the *SC*1 and *SC*2 components and assigned to the macromolecular constituents of the muscle tissue. The variability of muscle composition between the four groups and the two muscles studied was further investigated by statistically analysing the weights *c*1 and *c*2 (see Equation (1)), as explained in the sequel.

### 4.3. Histochemical Analysis

For histochemical analysis, we used serial transverse 10 μm thick cryosections of the skeletal muscles cut with Leica CM 1950 (Leica Microsystems, Germany). Semiquantitative lipid content analysis was adopted from Masgrau et al. [12]. Lipid staining was performed with Sudan black B powder which stains neutral lipids black as described by Goodpasteur et al. [92]. Semiquantitative glycogen content analysis was adopted from Fairchild et al. [93]. Polysaccharide staining was performed using Periodic acid–Schiff (PAS) staining procedure by McManus [94]. Semiquantitative collagen content analysis was adopted from Vesentini et al. [95]. Collagen bundles in tissue sections were stained using Sirius red (0.1% of Sirius red in saturated aqueous picric acid) as described by Junqueira et al. [96].

The slides were examined, and images were captured using a Nikon Eclipse 80i microscope (20× objective, numerical aperture: 0.50) equipped with a Nikon digitalized camera DS-Fi1 and computer software for image acquisition (NIS Elements Imaging software, version 3.22.15, Nikon, Tokyo, Japan). Tissue sections were captured at 2560 × 1920 pixel resolution, with a minimum of three fields of view randomly sampled for each muscle section. All images of similarly stained sections were captured using the same settings. The lipid content was analysed using the Ellipse software (ViDiTo, Kosice, Slovakia). The lipid content was estimated using the lipid content index, which was calculated as 100 times the ratio of the area of skeletal muscle tissue occupied by lipid droplets to the cross-sectional area of skeletal muscle tissue [12,92]. The glycogen content was analysed using ImageJ software [97]. The intensity of staining was evaluated in the red channel obtained using the colour deconvolution plugin [98]. The red channel was converted into grayscale 8-bit picture, and the mean grey value was measured. The collagen content was analysed using ImageJ and the Ellipse software. Colour deconvolution plugin and thresholding were used to segment the red-stained tissue (collagen) [98]. The abundance of collagen was expressed as collagen content index calculated as 100 times the ratio of the area of skeletal muscle tissue stained for collagen to the cross-sectional area of skeletal muscle tissue [95]. All study protocols were performed in a blind manner throughout the study.

### 4.4. Statistical Analysis

The GraphPad Prism 8.0 (GraphPad Software, San Diego, CA, USA) was used for all statistical analyses and graphing. The normality of all data in statistical analysis was analysed with the Shapiro–Wilk test. A repeated measure two-way ANOVA with Sidak post hoc tests was used to compare serial glucose measurements during OGTT. For comparison of animal weight and fasting glucose concentrations between study groups, a one-way ANOVA with Holm–Sidak post hoc test was used. To compare the histochemical indices of skeletal muscle composition between gluteus maximus and gastrocnemius muscles of the same animals and between study groups, the repeated measures two-way ANOVA with Sidak post hoc test for multiple comparisons was used. Statistical analysis of the FTIR spectroscopy results was based on comparison of the weight *c*1 between the two muscles within the group and between the study groups, and separately for the weight *c*2. Again, we used the repeated measures two-way ANOVA with Sidak post hoc test. It is important to mention that the unequal sample sizes of *c*1 and *c*2 were accounted for in ANOVA because we had to remove 1–3 outliers from certain samples determined by the Tukey fence method to satisfy the normality condition.

## 5. Conclusions

In this study, we analysed the composition of non-weight-bearing and weight-bearing skeletal muscle to detect modifications in tissue composition in old, obese, insulin-resistant, and STZ-induced diabetic mice and their respective control groups. Using FTIR spectroscopy independently of histochemical analysis, we found that the most pronounced effects on skeletal muscle composition occurred in the young STZ-induced diabetic group. Furthermore, we confirmed that the phenotype of skeletal muscle composition was different in the type 1 and type 2 DM models and that, in general, changes in skeletal muscle composition differed markedly according to muscle function typology. We showed that FTIR spectroscopy is a very convenient and valuable approach to detect relatively small variations in the macromolecular composition of skeletal muscle. To this end, an important step in FTIR analysis is the decomposition of spectra into independent spectral components defined over the entire frequency range and determined by the positive spectral bands reflecting the intrinsic vibration of specific macromolecules. Our results suggest that this spectroscopic method has great potential to complement or even replace various complex histochemical techniques for the early detection of metabolic abnormalities in skeletal muscle, disease staging, phenotypic classification, prognosis, and monitoring of disease progression associated with impaired insulin function.

## Figures and Tables

**Figure 1 ijms-23-12498-f001:**
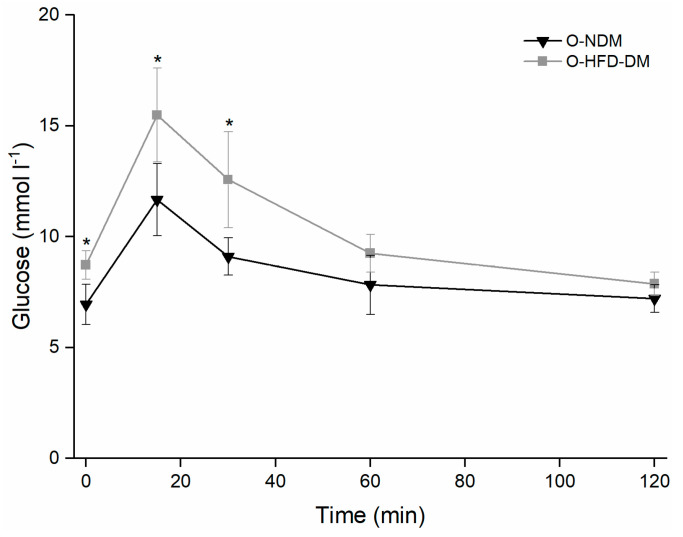
Oral glucose tolerance test measurements of 54-week-old non-diabetic mice (▼) (*n*  =  9) and 54-week-old high-fat diet-induced diabetic mice (■) (*n*  =  9). Data are means and standard deviation (SD); * *p * <  0.05 vs. O-NMD mice.

**Figure 2 ijms-23-12498-f002:**
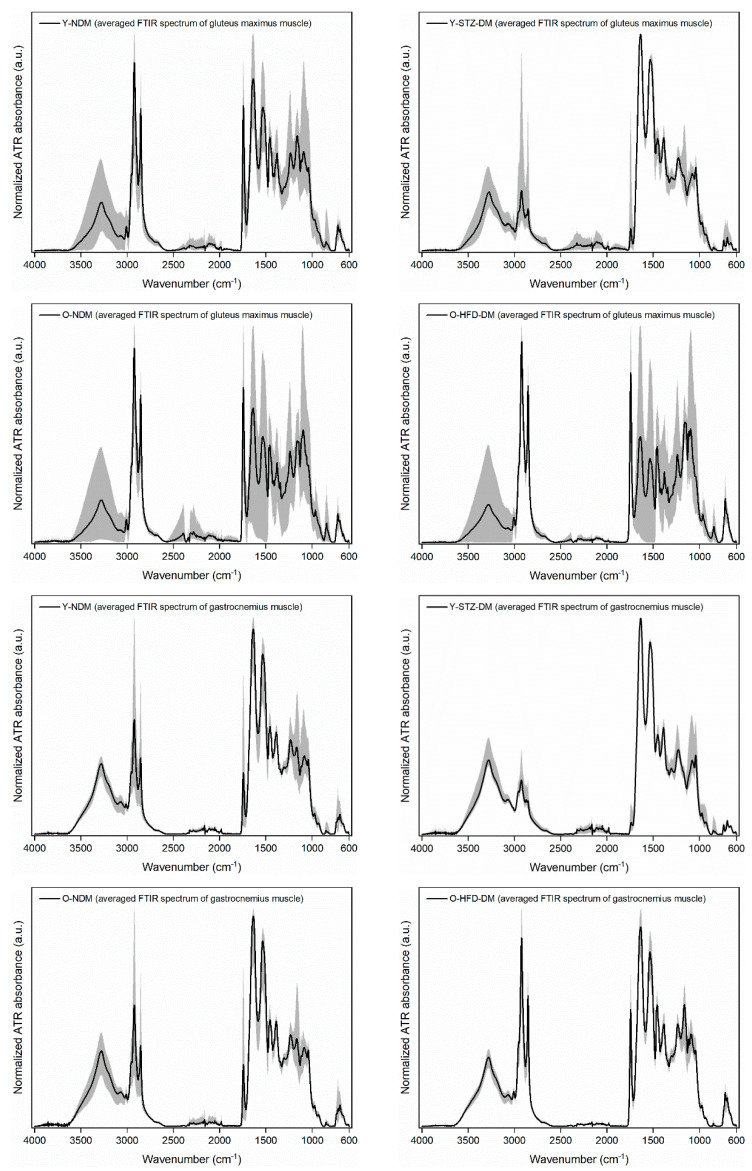
Averaged normalized ATR-FTIR spectra for gluteus maximus and gastrocnemius muscles of 13-week-old non-diabetic mice (*n* = 9) (Y-NDM), 13-week-old STZ-induced diabetic mice (*n* = 9) (Y-STZ-DM), 54-week-old non-diabetic mice (*n* = 9) (O-NDM), and 54-week-old HFD-induced obese insulin-resistant mice (*n* = 9) (O-HFD-DM), where the grey shaded area indicates the range between the minimum and maximum intensity at the corresponding wavenumber.

**Figure 3 ijms-23-12498-f003:**
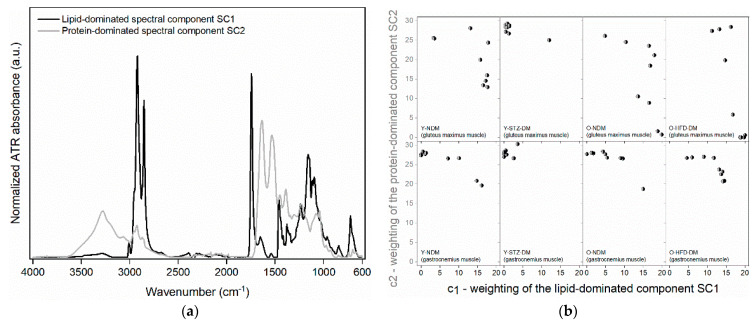
Two spectral components, *SC*1 and *SC*2, from the MCR decomposition (**a**) and the corresponding weights, *c*1 and *c*2, for the two spectral components obtained for gluteus maximus and gastrocnemius muscles (**b**) of 13-week-old non-diabetic mice (*n* = 9) (Y-NDM), 13-week-old STZ-induced diabetic mice (*n* = 9) (Y-STZ-DM), 54-week-old non-diabetic mice (*n* = 9) (O-NDM), and 54-week-old HFD-induced obese insulin-resistant mice (*n* = 9) (O-HFD-DM).

**Figure 4 ijms-23-12498-f004:**
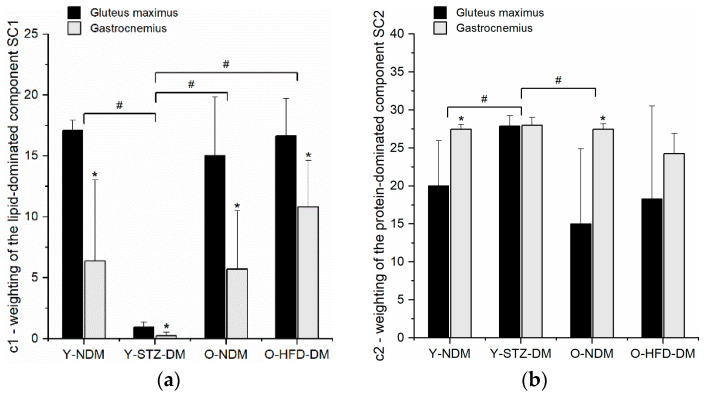
The values of *c*1 (**a**) and *c*2 (**b**) for the gluteus maximus and gastrocnemius muscles of 13-week-old non-diabetic mice (*n* = 9) (Y-NDM), 13-week-old STZ-induced diabetic mice (*n* = 9) (Y-STZ-DM), 54-week-old non-diabetic mice (*n* = 9) (O-NDM), and 54-week-old HFD-induced obese insulin-resistant mice (*n* = 9) (O-HFD-DM). Data are means and standard deviation (SD); ^#^
*p* < 0.05 for comparison between study groups and * *p*  <  0.05 for comparison of gluteus maximus vs. gastrocnemius in the same study group (two-way ANOVA with repeated measures).

**Figure 5 ijms-23-12498-f005:**
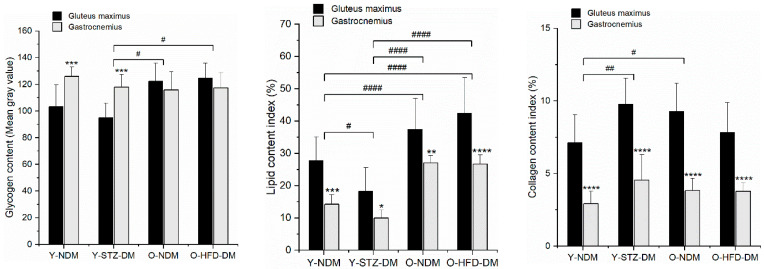
Glycogen, lipid, and collagen compositional changes determined by histochemical analysis of the gluteus maximus and gastrocnemius muscles of 13-week-old non-diabetic mice (*n* = 9) (Y-NDM), 13-week-old STZ-induced diabetic mice (*n* = 9) (Y-STZ-DM), 54-week-old non-diabetic mice (*n* = 9) (O-NDM), and 54-week-old HFD-induced obese insulin-resistant mice (*n* = 9) (O-HFD-DM). Data are means and standard deviation (SD); ^#^
*p* < 0.05, ^##^
*p* < 0.01, ^####^
*p* < 0.0001 for comparison between study groups and * *p*  <  0.05, ** *p* < 0.01, *** *p* < 0.001, **** *p* < 0.0001 for comparison of gluteus maximus vs. gastrocnemius in the same study group (two-way ANOVA with repeated measures).

**Figure 6 ijms-23-12498-f006:**
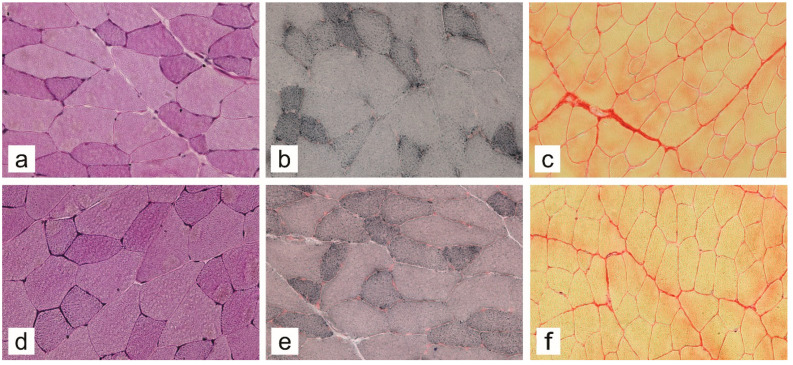
Representative images of PAS (glycogen) (**a**,**d**), Sudan Black B (lipids) (**b**,**e**), and Sirius red (collagen) (**c**,**f**), stained transverse sections of gluteus maximus (**a**–**c**) and gastrocnemius (**d**–**f**) muscles of 13-week-old non-diabetic mice. Magnification: 200×.

**Figure 7 ijms-23-12498-f007:**
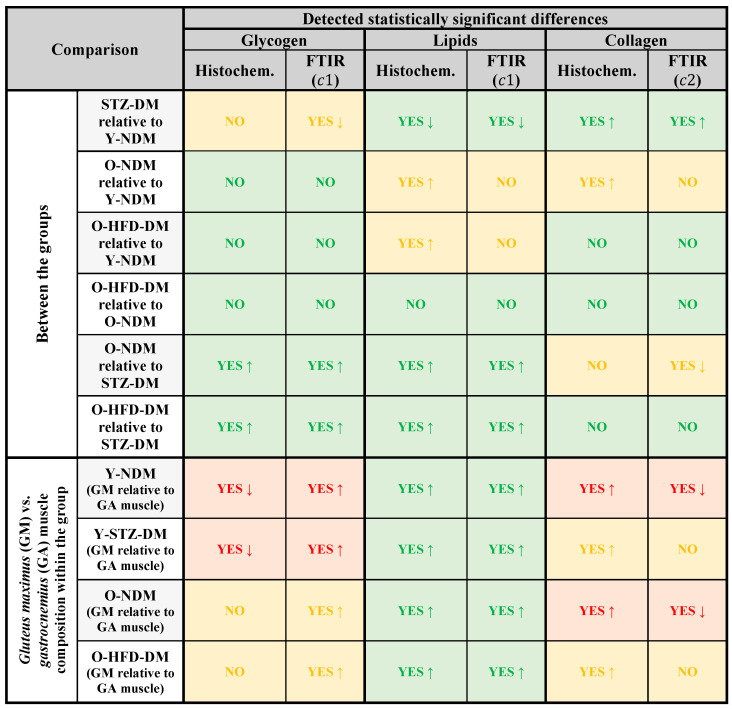
Comparability of FTIR spectroscopy and histochemical methods for detecting statistical differences in skeletal muscle tissue composition in 13-week-old non-diabetic mice (Y-NDM), 13-week-old STZ-induced diabetic mice (Y-STZ-DM), 54-week-old non-diabetic mice (O-NDM), and 54-week-old HFD-induced obese insulin-resistant mice (O-HFD-DM). Green boxes show the complete agreement of histochemical methods and FTIR in detecting statistical differences (YES—a statistical difference detected, NO—a statistical difference not detected) and the trend of the differences (↑ increasing trend and ↓ decreasing trend relative to the reference). Yellow boxes show a disagreement in the detection of statistical differences. Red boxes show agreement in detected statistical differences but disagreement in the trend.

**Table 1 ijms-23-12498-t001:** Body mass and fasting glucose concentrations of 13-week-old non-diabetic mice (*n* = 9) (young non-diabetic mice, Y-NDM), 13-week-old STZ-induced diabetic mice (*n* = 9) (Y-STZ-DM), 54-week-old old (O) non-diabetic mice (*n* = 9) (O-NDM), and 54-week-old HFD-induced obese insulin-resistant mice (*n* = 9) (O-HFD-DM).

	Y-NMD	Y-STZ-DM	O-NDM	O-HFD-DM
Body mass (g)	26.2 ± 1.3	19.9 ± 1.7 *	26.7 ± 2.5	36.9 ± 2.2 *
Fasting glucose (mmol L^−1^)	6.7 ± 1.0	31.3 ± 2.8 ^#^	6.9 ± 0.9	8.7 ± 0.7 ^Ψ^

Data are means ± standard deviation (SD); * *p* < 0.0001 vs. Y-NMD and O-NMD mice, ^#^
*p* < 0.0001 vs. Y-NMD, O-NMD, and O-HFD-DM, ^Ψ^
*p* < 0.05 vs. Y-NMD and O-NMD.

**Table 2 ijms-23-12498-t002:** Band assignments of the two spectral components, *SC*1 and *SC*2, obtained from the MCR decomposition of the ATR-FTIR spectra.

*SC*1—Peak Wavenumber (cm^−1^)	*SC*1—Spectral Assignment	*SC*2—Peak Wavenumber (cm^−1^)	*SC*2—Spectral Assignment
**3467** **3290**	O-H stretching of carbohydrates [21]	**3307** **3294** **3272** **3067**	amide A and amide B band originating from a Fermi resonance between the N-H stretching vibration and the first overtone of amide II ([22,23,24,25,26] and references therein)
**3007**	Olefinic C=CH stretching vibration of unsaturated fatty acids ([18,26] and references therein, [27])		
**(2959 shoulder)** **2922** **(2874 shoulder)** **2853**	CH_3_ and CH_2_ symmetric and antisymmetric stretching mostly of fatty acids and phospholipids [22,27,28,29,30]	**2960** **2924** **2874** **(2853 shoulder)**	CH_3_ and CH_2_ symmetric and antisymmetric stretching mostly of protein side chains [22,31]
**1743**	C=O stretching of lipid esters [20,27,32]		
		**1693**	amide I (C=O stretching, C-N stretching, CNN deformation) of β-sheet protein secondary structures [33,34]
		**1679**	amide I (C=O stretching, C-N stretching, CNN deformation) of β-sheet protein secondary structures [34,35]
**1653**	amide I (C=O stretching, C-N stretching, CNN deformation) of α-helical protein secondary structures [32,34,35,36,37]	**1652**	amide I (C=O stretching, C-N stretching, CNN deformation) of α-helical protein secondary structures [32,34,35,36,37]
**1629**	amide I (C=O stretching, C-N stretching, CNN deformation) of β-sheet protein secondary structures [28,32,34,38,39]	**1629**	amide I (C=O stretching, C-N stretching, CNN deformation) of β-sheet protein secondary structures [28,32,34,38,39]
		**1568**	amide II (C-N stretching coupled with N-H bending) of β-turn protein secondary structures [35]
**1542**	amide II (C-N stretching coupled with N-H bending) of α-helical and β-sheet protein secondary structures [28,36,37]	**1547**	amide II (C-N stretching coupled with N-H bending) of α-helical and β-sheet protein secondary structures [28,36,37]
		**1530**	amide II (C-N stretching coupled with N-H bending) of β-turn protein secondary structures [35]
		**1513**	vibration of the tyrosine ring in proteins [28,29,40]
**1464** **1457** **1436** **1417**	CH_3_ and CH_2_ bending vibrations mostly of fatty acids and phospholipids [27,28,41,42,43,44] and cis =C–H bending at 1417 cm^−1^ of unsaturated fatty acids [45]	**1468** **1454** **1420** **1387**	CH_3_ and CH_2_ bending vibrations mostly of protein side chains [28,42,44,46,47,48]
**1401**	C=O symmetric stretching of COO^−^ groups of fatty acids [49,50,51]		
**1377** **1361**	CH_3_ symmetric bending mostly of fatty acids and phospholipids [27,43,52]		
**1343** **1315**	CH_3_ wagging mostly of fatty acids and phospholipids [43,51]	**1340**	CH_2_ side chain vibration in collagen ([16,48,53] and references therein)
		**1309**	amide III of α-helical protein secondary structures [54]
**1301**	in-phase CH_2_ twist mode of fatty acids [55]		
**(1279** **1263 shoulders)** **1239**	PO_2_^−^ antisymmetric stretching of phospholipids [28,43]	**1241**	amide III vibration from C-N stretching, N-H bending vibration, and wagging vibration of CH_2_ groups in the glycine backbone and proline side chains of collagen ([16,20,32,53] and references therein) and PO_2_^−^ antisymmetric stretching of RNA ([56] and references therein)
		**1231**	PO_2_^−^ antisymmetric stretching of DNA ([56] and references therein)
		**1172**	vibrations of COH groups of serine, threonine, and tyrosine residues in proteins (1161 cm^−1^—hydrogen-bonded CO group, and 1173 cm^−1^—non-hydrogen-bonded CO groups) [57,58]
**1161** **1142**	C-O-C bonds between the glycerol carbon and fatty acid ester carbon of triglycerides [27,59]		
		**1122**	symmetric phosphodiester stretching band mainly of RNA ([60] and references therein)
		**1103**	PO_2_^−^ symmetric stretching and C-O stretching of deoxyribose [52]
**1117** **1096**	–C–O stretching of esters of fatty acids [27,45] and PO_2_^−^ symmetric stretching of phospholipids [43]		
**1082**	C-C stretching of glycogen [28,61,62]	**1081**	C-O and C-C stretching of glycated collagen [16,63] and PO_2_^−^ symmetric stretching of nucleic acids ([28,56] and references therein)
**1060**	C-O stretching of P-O-C of phospholipids [64]		
		**1041**	C-C-O and C-O-H bending of glycated collagen [16,63] and C-O stretching of RNA D-ribose (skeletal motions of nucleic acids) ([56] and references therein)
**1023**	C-O-H bending of glycogen [28,52,61,62]		
**987**	phospholipids [43] and =C-H bending of monosaccharides and polysaccharides [30]		
		**972** **(952 shoulder)** **929**	C-C stretching of the Z-DNA backbone [65,66] and Ribose phosphate main chain vibration of RNA backbone [65]
**964** **(947 shoulder)**	=C-H out-of-plane bending of unsaturated fatty acids [27,49] and possibly with contribution of N^+^-(CH_3_)_3_ vibration of phospholipids [43]		
**913** **891**	fatty acids [27] and phospholipids [43]		
		**849** **830**	tyrosine [67,68]
**843** **(857 shoulder)**	fatty acids [27] and phospholipids [43] and possibly with contribution of glycogen [21]		
**828**	P-O antisymmetric stretching of P-O-C of phospholipids [43]		
		**792**	guanine C3′-endo/syn conformation in the Z-DNA [69]
		**779 and 772**	guanine–uracil wobble base pair [70,71]
**767**	CH_2_ rocking of glycogen [21]		
**722** **(692 shoulder)**	CH_2_ rocking of unsaturated aliphatic chains [72] and =C–H out-of-plane bending of unsaturated fatty acids [27,73]	**739** **699** **661**	amide IV O=C–N deformation and amide V N–H out-of-plane deformation [23]

**Table 3 ijms-23-12498-t003:** Summary of statistically significant differences in muscle composition between 13-week-old non-diabetic mice (*n* = 9) (Y-NDM), 13-week-old STZ-induced diabetic mice (*n* = 9) (Y-STZ-DM), 54-week-old non-diabetic mice (*n* = 9) (O-NDM), and 54-week-old HFD-induced obese insulin-resistant mice (*n* = 9) (O-HFD-DM), and statistically significant differences in muscle composition between gluteus maximus and gastrocnemius muscles within each group at the 0.05 significance level based on the results shown in Figure 4 combined with the band assignments from Table 2.

Comparison		Studied Effect/ Purpose of Comparison	Statistically Significant Differences in Macromolecular Composition (Based on Results Shown in Figure 4 Combined with Band Assignments from Table 2)	
			*c*1	*c*2
**Between the groups**	**STZ-DM** **relative to** **Y-NDM**	Changes in the muscle composition in STZ-induced type 1 DM relative to the control group/possibility of differentiating the two groups.	Decreased *c*1 for the STZ-DM group compared to the Y-NDM group means **decreased**: - Unsaturated fatty acids - Fatty acid esters (mostly triglycerides) - Phospholipids - Glycogen	Increased *c*2 for the STZ-DM group compared to the Y-NDM group means **increased**: - Proteins in general - Collagen - Glycated collagen - Tyrosine - Nucleic acids
			Changed overall protein composition patterns.	
	**O-NDM** **relative to** **Y-NDM**	Changes in the muscle composition in relation to age/possibility of differentiating the two groups.	-	-
	**O-HFD-DM** **relative to** **Y-NDM**	Changes in the muscle composition in relation to age and obesity with insulin resistance.	-	-
	**O-HFD-DM** **relative to** **O-NDM**	Changes in the muscle composition in relation to obesity with insulin resistance relative to the control group/possibility of differentiating the two groups.	-	-
	**O-NDM** **relative to** **STZ-DM**	Differentiation between muscle composition affected by age and muscle composition affected by STZ-induced type 1 DM/possibility of differentiating the two groups.	Increased *c*1 for the O-NDM group compared to the STZ-DM group means **increased**: - Unsaturated fatty acids - Fatty acid esters (mostly triglycerides) - Phospholipids - Glycogen	Decreased *c*2 for the O-NDM group compared to the STZ-DM group means **decreased**: - Proteins in general - Collagen - Glycated collagen - Tyrosine - Nucleic acids
			Changed overall protein composition patterns.	
	**O-HFD-DM** **relative to** **STZ-DM**	Differentiation between muscle composition affected by age and obesity with insulin resistance factors, and muscle composition affected by STZ-induced type 1 DM/possibility of differentiating muscle samples based on FTIR spectrum.	Increased *c*1 for the O-HFD-DM group compared to the STZ-DM group means **increased**: - Unsaturated fatty acids - Fatty acid esters (mostly triglycerides) - Phospholipids - Glycogen	-
			Changed overall protein composition patterns.	
** *Gluteus maximus* ** **(GM) vs. *gastrocnemius* (GA) muscle composition within the group**	**Y-NDM** **(GM relative to GA muscle)**	Differentiation between the composition of the gluteus maximus and the gastrocnemius muscle at a younger age/possibility of differentiating the non-weight-bearing and weight-bearing muscles.	Increased *c*1 for the GM muscle compared to the GA muscle of the Y-NDM group means **increased**: - Unsaturated fatty acids - Fatty acid esters (mostly triglycerides) - Phospholipids - Glycogen	Decreased *c*1 for the GM muscle compared to the GA muscle of the Y-NDM group means **decreased**: - Proteins in general - Collagen - Glycated collagen - Tyrosine - Nucleic acids
			Changed overall protein composition patterns.	
	**Y-STZ-DM** **(GM relative to GA muscle)**	Differentiation between the composition of the gluteus maximus and the gastrocnemius muscle affected by STZ-induced type 1/possibility of differentiating the non-weight-bearing and weight-bearing muscles.	Increased *c*1 for the GM muscle compared to the GA muscle of the Y-STZ-DM group means **increased**: - Unsaturated fatty acids - Fatty acid esters (mostly triglycerides) - Phospholipids - Glycogen	
			Changed overall protein composition patterns.	
	**O-NDM** **(GM relative to GA muscle)**	Differentiation between the composition of the gluteus maximus and the gastrocnemius muscle in older age/possibility of differentiating the non-weight-bearing and weight-bearing muscles.	Increased *c*1 for the GM muscle compared to the GA muscle of the O-NDM group means **increased**: - Unsaturated fatty acids - Fatty acid esters (mostly triglycerides) - Phospholipids - Glycogen	Decreased *c*2 for the GM muscle compared to the GA muscle of the O-NDM group means **decreased**: - Proteins in general - Collagen - Glycated collagen - Tyrosine - Nucleic acids
			Changed overall protein composition patterns.	
	**O-HFD-DM** **(GM relative to GA muscle)**	Differentiation between the composition of the gluteus maximus and the gastrocnemius muscle affected by age and obesity with insulin resistance factors/possibility of differentiating the non-weight-bearing and weight-bearing muscles.	Increased *c*1 for the GM muscle compared to the GA muscle of the O-HFD-DM group means **increased**: - Unsaturated fatty acids - Fatty acid esters (mostly triglycerides) - Phospholipids - Glycogen	
			Changed overall protein composition patterns.	

## Data Availability

The data presented in this study are available upon request from the corresponding author.

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
