# Peer review of "Application of FTIR Spectroscopy to Detect Changes in Skeletal Muscle Composition Due to Obesity with Insulin Resistance and STZ-Induced Diabetes"

_ijms, 2022, doi:10.3390/ijms232012498_

Round 1

Reviewer 1 Report

The work presented here describes an application of FTIR to analyze samples from two types of mouse muscle tissue and aims to identify differences in these tissues in function of diabetes, age and obesity.

The work is relevant and demonstrates another application of this type of vibrational spectroscopy. 

The introduction is adequate and the objectives are clear.

Although the authors do not mention the acquisition of sample replicates, the experimental design seems adequate. The 2cm-1 resolution may have contributed to a lower reproducibility between samples.

The analysis by MCR-ALS, despite the variability between samples, allows the identification, in general, of the contribution of the lipid and protein fractions in each of the tissues studied. 

In Table 2 what is the benefit of separately assigning the sc1 and sc2 components?

The passage of the results from Figure 4 to Table 3, in my opinion, is unclear. The reader does not have access to how the results of figure 4 are translated into table 3 and the information is difficult to understand.

Table 3 is uninformative and confusing and will need to be completely reformulated. To reformulate the table I suggest additional analysis in order to introduce more relevant and accurate results. For this the authors can apply other multivariate analysis tools, a simple PCA or PCA-DA, will serve to identify concrete differences between samples, this comparison can be applied to sets of 3 samples.  

Reference 65 can be replaced by a scientific article.

Author Response

Dear reviewer,

Thank you for reviewing our manuscript IJMS-1951340, “Application of FTIR spectroscopy to detect changes in skeletal muscle composition due to obesity with insulin resistance and STZ-induced diabetes”. Below you will find your comments with our responses in blue font.

Reviewer’s comments and authors’ responses

The work presented here describes an application of FTIR to analyze samples from two types of mouse muscle tissue and aims to identify differences in these tissues in the function of diabetes, age, and obesity.

The work is relevant and demonstrates another application of this type of vibrational spectroscopy.

The introduction is adequate and the objectives are clear.

Although the authors do not mention the acquisition of sample replicates, the experimental design seems adequate. The 2 cm-1 resolution may have contributed to a lower reproducibility between samples.

The analysis by MCR-ALS, despite the variability between samples, allows the identification, in general, of the contribution of the lipid and protein fractions in each of the tissues studied.

  1. In Table 2 what is the benefit of separately assigning the SC1 and SC2 components?

Response: The main purpose of our decomposition of the spectra into the spectral components was to more easily identify the portion of a particular group of macromolecules in the tissue based on the characteristic absorption bands. Since the spectral component SC1 from the decomposition consists mainly of the absorption bands characteristic of unsaturated fatty acids, fatty acid esters, phospholipids, and glycogen, this means that the portion of the component SC1 in the FTIR-ATR absorption spectrum, expressed by weight c1, indicates the portion of these specific constituents. Similarly, SC2 consists mainly of the absorption bands characteristic to proteins in general, collagen, glycated collagen, tyrosine, and nucleic acids, which means that the portion of the SC2 component contributing to the ATR-FTIR absorption spectrum, expressed by weight c2, indicates the portion of these specific constituents. To explain more clearly the purpose of the spectral decomposition and band assignment, we have modified the paragraph following directly after Table 2. It is highlighted in yellow in the revised manuscript.

  1. The passage of the results from Figure 4 to Table 3, in my opinion, is unclear. The reader does not have access to how the results of figure 4 are translated into Table 3 and the information is difficult to understand. Table 3 is uninformative and confusing and will need to be completely reformulated. To reformulate the table I suggest additional analysis in order to introduce more relevant and accurate results. For this the authors can apply other multivariate analysis tools, a simple PCA or PCA-DA, will serve to identify concrete differences between samples, this comparison can be applied to sets of 3 samples.

Response: We appreciate your comment. Accordingly, we have resolved the ambiguities associated with to the findings presented in Table 3 of the original manuscript by rewording the paragraph following directly after Table 2, the paragraph following Figure 4, and the text in Table 3. These text changes are highlighted in yellow in the revised manuscript.

We believe that a more detailed explanation of how we obtained the information presented in Table 3 further justifies our multivariate approach via the MCR-ALS decomposition into the two spectral components and that the additional multivariate analysis is not necessary.

  1. Reference 65 can be replaced by a scientific article.

Response: We have replaced reference 65 of the original manuscript with reference “Stani, C.; Vaccari, L.; Mitri, E.; Birarda, G. FTIR Investigation of the Secondary Structure of Type I Collagen: New Insight into the Amide III Band. Spectrochim. Acta - Part A Mol. Biomol. Spectrosc. 2020, 229, 118006, doi:10.1016/j.saa.2019.118006” in revised manuscript.

Reviewer 2 Report

-Application of FTIR in detecting changes in macromolecular level due to DM type 2 is a pioneering study.

-In the result section (page 14) in addition to Bozkurt et al. (15), discussion on reference 19 is recommended where the alteration of different macromolecular components was studied in DM type 1 affected rat heart.

-Samples used in the study had type 2 pre-diabetic states, future studies are recommended with samples with advanced DM type 2.

Author Response

Dear reviewer,

Thank you for reviewing our manuscript IJMS-1951340, “Application of FTIR spectroscopy to detect changes in skeletal muscle composition due to obesity with insulin resistance and STZ-induced diabetes”. Below you will find your comments with our responses in blue font.

Reviewer’s comments and authors’ responses

The application of FTIR in detecting changes in macromolecular level due to DM type 2 is a pioneering study.

  1. In the result section (page 14) in addition to Bozkurt et al. (15), a discussion on reference 19 is recommended where the alteration of different macromolecular components was studied in DM type 1 affected rat heart.

Response: We have followed your recommendation and added a short note to the first paragraph after Table 3, referring to reference (19). The text is highlighted in yellow in the revised manuscript.

  1. Samples used in the study had type 2 pre-diabetic states, future studies are recommended with samples with advanced DM type 2.

Response: Thank you for your suggestion. We will take it into consideration in our further studies.

Round 2

Reviewer 1 Report

After revision the paper is ready to be published.